# Effects of Electrospun Potato Protein–Maltodextrin Mixtures and Thermal Glycation on Trypsin Inhibitor Activity

**DOI:** 10.3390/foods11070918

**Published:** 2022-03-23

**Authors:** Monika Gibis, Franziska Pribek, Jochen Weiss

**Affiliations:** Department of Food Material Science, Institute of Food Science and Biotechnology, University of Hohenheim, Garbenstr. 25, 70599 Stuttgart, Germany; franziskapribek@web.de (F.P.); j.weiss@uni-hohenheim.de (J.W.)

**Keywords:** glycoconjugates, potato protein, maltodextrin, trypsin inhibitor, free lysine groups, needleless electrospinning, glycation

## Abstract

Fibers of potato protein and polysaccharides were obtained by needleless electrospinning. Mixtures of maltodextrin DE2 (dextrose equivalent) (0.8 g/mL), DE21 (0.1 g/mL), and different concentrations of potato protein (0.05, 0.1, 0.15, and 0.2 g/mL) were used for fiber production. Glycation was performed via the Maillard reaction after thermal treatment (0/6/12/24/48 h, 65 °C, 75% relative humidity). The effects of electrospinning and heating on trypsin inhibitor activity (IA) were studied. The results of the IA assay showed that electrospinning and glycation caused significant differences in IA among blends, heating times, and the interaction of blend and heating time (*p* < 0.001). The higher the protein content in the fibers, the higher the IA. The lowest IA was found in the mixture with the lowest protein content after 48 h. In other blends, the minimum IAs were found between 6 and 12 h of heating. The determination of the free lysine groups showed a nonsignificant decrease after heating. However, higher free lysine groups per protein (6.3–9.5 g/100 g) were found in unheated fibers than in the potato protein isolate (6.0 ± 0.5 g/100 g). The amide I and amide II regions, detected by the Fourier transform infrared spectra, showed only a slight shift after heating.

## 1. Introduction

Protein isolates are used in the production of a wide range of foods [1,2]. In addition to their nutritional value, these isolates are particularly important for their physicochemical properties, such as emulsifying, gelling, and foaming properties, or entrapment of bio-actives; in particular, the reuse of agro-food waste byproducts is of importance [3,4]. Due to the increasing demand for alternatives to animal proteins, plant proteins, such as potato protein (PP), are becoming increasingly important [5]. The protein fraction of potato protein can be obtained from the residual streams of starch production, known as potato fruit juice [6,7]. In addition to about 2% protein, potato fruit juice consists of water, amino acids, and starch [8]. The protein fraction of the potato can be divided into three groups: patatin, protease inhibitors, and high-molecular weight proteins [9]. Patatin accounts for about 40% of the potato’s soluble protein, while protease inhibitors account for 30–40% [10,11,12]. The potato may contain up to 13 inhibitor species that can inhibit some proteases, including trypsin, pancreatic peptidase E, and chymotrypsin A, thereby affecting the digestibility and biological value of the protein [13,14]. The serine protease inhibitors account for the largest proportion of potato inhibitors, approximately 20%, and inhibit trypsin and chymotrypsin [11,12]. Potato serine proteases inhibitors (PSPI) are double-headed heterodimeric Kunitz-type inhibitors [15]. By steric hindrance, they block the access of substrates to the catalytic centers of proteinases, mimicking the substrate of the enzyme with their reactive loops [15,16]. In addition to their biological value as suppliers of essential amino acids, potato protein isolates also have useful physicochemical properties, such as high solubility [17], emulsifying, and foaming abilities, and may, therefore, have the potential to be used as a food ingredient [3,18]. Natural potato protein shows good functional properties, such as binding fat and water, but, due to the presence of protease inhibitors, its nutritional value is unfavorable. Studies have shown that glycation of potato protein can improve the applicability of these proteins. Potato glycoproteins, when glycated with galactose or galactooligosaccharides, have been shown to have higher heat stability, stability to low pH, and a high antioxidant activity [19].

The Maillard reaction involves a series of complex chemical reactions between the amino group of an amino acid, protein, or peptide, and the carbonyl group of a reducing sugar [18,20]. The Maillard reaction can be divided into initial, intermediate, and final stages [21]. The glycation is based on the Amadori rearrangement in the first stage of the Maillard reaction [22,23]. By glycation via the Maillard reaction, conjugates of proteins with polysaccharides can be obtained [3,18]. The conjugation can be carried out under wet or dry conditions, or by electrospinning prior to a thermal treatment [3,18]. Electrospinning is a simple and inexpensive method for the production of nanofibers, in which jets are generated from a charged polymer solution using high voltage [24,25]. The fibers are then collected on a grounded collector [24]. Electrospinning has some advantages over other possible method; by producing fine fibers with a small diameter, large surface area, and close contact between the individual molecules in the fibers, glycation is supposed to proceed faster, more efficiently, and more economically [26,27]. Some applications in the food sector include active packaging with electrospun fibers that contain bioactives, such as antioxidants and antimicrobials [28,29]. Some studies have already addressed the possibility of glycation of electrospun fibers prepared from whey, soy, or pea protein with dextrans or maltodextrins [5,26,27,30,31]. The preparation of electrospun potato protein-maltodextrin fibers followed by thermal treatment was presented in a recent study [32]. The formation of conjugates between proteins and polysaccharides can not only lead to improved functional properties of the proteins, but also to structural changes, which can influence their biological activity [33]. One study suggested that an amino–carbonyl reaction of trypsin inhibitors (partially derived from soybeans) with glucose resulted in a reduction of trypsin inhibitory activity [34].

The aim of this work was to study the trypsin inhibitory activity of the potato protein after electrospinning and glycation with maltodextrin via the Maillard reaction. As a hypothesis, we postulated that the combined treatments would result in a decrease in trypsin inhibitory activity as a result of the large surface area and mild glycation conditions. Furthermore, the effects of heating time on the number of available lysine groups were investigated. FTIR spectra were used to provide information about the chemical structure and the progress of the Maillard reaction in the samples.

## 2. Materials and Methods

### 2.1. Materials

Maltodextrin, with a dextrose equivalent (DE) of 2 and a molecular weight Mw of ~124 kDa (MD DE2) (ElianeTM MD2 by Avebe, Veendam, The Netherlands), maltodextrin DE 21 (Glucidex^®^ 21D by Roquette, Lestrem, France) (MD DE21), and potato protein (PP) (Solanic^®^ 300 by Avebe, Veendam, The Netherlands), with a protein content of 91.16 ± 0.3 g/100 g, were used. The chemicals ortho-phtaldialdehyde (OPA) (purity ≥ 99%), calcium chloride CaCl_2_ (≥98%), trichloroacetic acid (≥99%), and sodium dodecyl sulfate (SDS) (≥99%) were purchased from Carl Roth GmbH & Co. KG, (Karlsruhe, Germany), along with TRIS-buffer from AppliChem GmbH (Darmstadt, Germany), and ethanol (≥99.8%), β-mercaptoethanol (≥98%), L-lysine, azocasein, bovine pancreatic trypsin, sodium hydroxide (NaOH), and hydrochloric acid (HCl) from Sigma-Aldrich Chemie GmbH (Steinheim, Germany), as well as 0.1 M sodium tetraborate buffer (pH 9.7) from Merck KGaA (Darmstadt, Germany).

### 2.2. Preparation of Electrospun Fibers and Their Thermal Treatment

The electrospun fibers were prepared as recently described in [32]. Each solution was prepared with 80 g maltodextrin MD DE2 and 10 g maltodextrin MD DE21, and an addition of potato protein powder of 5, 10, 15, or 20 g, respectively. The components were first mixed together dry; then, 100 mL double distilled water was added. The following ratios were achieved: blends A (80:10:5), B (80:10:10), C (80:10:15), and D (80:10:20). The spinning solutions were stirred overnight at ambient temperature (100 rpm).

After stirring, the blend solution was degassed under vacuum (~600 mbar, 25 °C) for 30 min and electrospun by needleless electrospinning [5,32]. The electrospinning equipment (Technical workshop of the University of Hohenheim) contained a spinneret block, a grounded collector, a command block, and a high voltage supply (SL 60, Spellman, Hauppauge, NY, USA). The parameters (applied voltage 64 kV) were needle-target distance 15.6 cm, rotating stainless-steel cylinder (140 rpm), and collector cylinder rotating (30 rpm)), and the environmental conditions were 22 °C and a relative humidity (RH) of 15–25%. Each blend of the electrospun fibers was subjected to a thermal treatment (65 °C and 75% RH for 0, 6, 12, 24, and 48 h in a climate cabinet (HCP 50, Memmert GmbH + Co. KG, 91126, Schwabach, Germany). The fiber production rate was 3–6 g/h [32]. The heated and unheated electrospun fibers were finely ground into powder for further analysis [32].

### 2.3. Determination of Available Lysine Groups

The content of available lysine groups was determined using the ortho-phtaldialdehyde (OPA) method. OPA forms a fluorescent complex with lysine, which can be analyzed fluorometrically. The OPA assay was performed as described in [35]. The powdered samples were diluted in double distilled water to a protein concentration of approximately 15 mg/mL. Then, 25 µL of the diluted sample was mixed with 475 µL double distilled water and 500 µL 12% *w*/*v* SDS. As a negative control, 500 µL of double distilled water was mixed with 500 µL 12% *w*/*v* SDS. All samples were stored overnight at 4 °C. The OPA reagent was freshly prepared on the day of analysis. For 55 mL OPA reagent, 80 mg OPA was dissolved in 2 mL ethanol in an ultrasonic bath. Then, 50 mL 0.1 M sodium tetraborate buffer (pH 9.7), 5 mL of a 20% *w*/*v* SDS solution, and 0.2 mL of β-mercaptoethanol were added. Next, 8 µL of the sample/blank was pipetted into a 96-well black microtiter plate (Brand GmbH & Co. KG, Wertheim, Germany), with 8 µL 0.1 M sodium tetraborate buffer (pH 9) and 250 µL OPA reagent. After 2 min of shaking at 25 °C, the plate was read on a microplate reader (BioTek Instruments GmbH, Bad Friedrichshall, Germany) at λ_ex_ = 340 nm and λ_em_ = 455 nm. Triplicates of each sample were prepared, which were then pipetted onto the plates three times. L-lysine, with concentrations of 1, 0.8, 0.6, 0.4, 0.2, and 0.1 mg/mL, was used to create a calibration curve to quantify the available lysine groups.

### 2.4. Total Protein Content

The total nitrogen contents of the unheated and heated fibers were determined according to Dumas (AS § 64 L 06.00-20, BVL [36]) using the following device: DUMATHERM^®^ (C. Gerhardt GmbH & Co. KG, Königswinter, Germany). The sample preparation was described analogously [32]. EDTA was used as a standard for calibration. The total protein content was determined by multiplying the measured nitrogen concentration by a nitrogen–protein conversion factor of 6.25 for potato protein [37].

### 2.5. Quantitative Analysis of Trypsin Inhibitor Activity

The assay for determining the trypsin inhibitor activity was performed according to Spelbrink et al. [38]. In brief, 30 g/L of azocasein stock solution was always freshly prepared on the day of analysis. Azocasein was dissolved in a 100 mM, pH 8.5 TRIS-buffer, containing 5 mM CaCl_2_, which was heated to 50 °C. The stock solution was then cooled down to 37 °C. The 0.3–0.4 mg/mL trypsin solution was prepared directly before the analysis by dissolving the bovine pancreatic trypsin in 1 mM HCl. For sample preparation, a sample stock solution was made by dissolving the powdered samples with acetic acid (pH 3.3) to obtain a protein content of approximately 1 mg/mL. The stock solution was then diluted to 1:10, 1:20, 1:50, 1:100, and 1:200 with double distilled water. Next, 125 µL from each sample dilution was mixed with either 25 µL trypsin stock solution or 25 µL water (as negative control) and 225 µL warm azocasein stock solution and was then incubated at 37 °C. After 30 min, 150 µL of 15% (*w*/*v*) trichloroacetic acid was added to stop the reaction. Then, the insoluble components were separated into a Z32 HK centrifuge (Hermle Labortechnik GmbH, Wehingen, Germany) at 15,000× *g* for 10 min. After centrifugation, 100 µL of the supernatants was pipetted into a microtiter plate together with 100 µL of 1.5 M NaOH solution. Each dilution was determined three times by photometric measurement. The absorbance was then measured at 450 nm on a microplate reader (BioTek Instruments GmbH, Bad Friedrichshall, Germany).

The amount of trypsin that was inactivated by a certain amount of the sample is called the inhibition activity (IA). Its calculation was derived from Spelbrink et al. [38], and is shown in Equation (1).
(1)$$\mathrm{IA}=\frac{-\mathrm{Slope}\cdot \mathrm{QTA}}{\left(\mathrm{PC}-\mathrm{NC}\right)}$$

IA is the inhibition activity (the slope of the absorption at 450 nm against the quantity of protein in the well of the microtiter plate), QTA is the quantity of trypsin in each well of the microtiter plate, PC is the positive control (the absorption without inhibitor present (the y axis intercept of the resulting line)), and NC is the negative control (the absorption without enzyme present).

### 2.6. Determination of Fiber Diameter

The fibers were investigated with a scanning electron microscope (SEM) (JSM-IT100 by JEOL, Freising, Germany) at an applied voltage of 2 kV under high vacuum. The average diameters were measured from 20 SEM images using ImageJ (National Institute of Health, Bethesda, MD, USA) as described in [32].

### 2.7. Determination of the Browning Index

Before grinding, the browning index of the unheated and heated fibers was measured with a Chromameter (CR-400/410 with data processor DP-400, Konia Minolta, Inc., Chiyoda, Japan). For calibration, a white standard tile (Y = 93.5, x = 0.3114, y = 0.3190) was used. The browning index was analyzed as described in [20,32] and calculated from the *L* a* b** values using Equations (2) and (3).
(2)$$x=\frac{a^{*}+1.75\;\cdot \;L^{*}}{5.645\;\cdot \;L^{*}+a^{*}-3.012\;\cdot b^{*}}$$
(3)$$\mathrm{BI}=\frac{100\;\left(x-0.31\right)}{0.172}$$

### 2.8. FTIR Analysis

The FTIR spectra of the heated and the unheated fibers were conducted using a Spectrum 100 (PerkinElmer, Beaconsfield, UK) equipped with a universal attenuated total reflectance (ATR-FTIR) accessory (PerkinElmer, Spectrum 100, Beaconsfield, UK). In total, 64 scans were measured per spectrum within the wavenumber range of 650 to 4000 cm^−1^, and then averaged using a resolution of 4 cm^−1^. The determination of peaks was performed with Spectrum 10 Spectroscopy Software (PerkinElmer, Beaconsfield, UK) and the peak deconvolution app (OriginLab Corporation, Northampton, MA, USA). The principal component analysis for the FTIR spectra was performed using the “Principal Component Analysis for Spectroscopy” app (OriginLab Corporation, Northampton, MA, USA).

### 2.9. Statistical Analysis

Means, standard deviations, and analyses of variance to determine significant differences (α = 0.05) were calculated with Excel (Microsoft, Redmond, WA, USA) and Sigma Plot 14 (Systat Software Inc., San Jose, CA, USA). As an assumption, the data were tested for normality and equality of variances (*p* < 0.05). A two-way ANOVA with post hoc test (Student-Newman-Keuls-test), one-way ANOVA, or one-way ANOVA on ranks (Kruskal-Wallis-test), when the test for normality or equal variance failed, was used to detect statistical differences (*p* < 0.05), which were labeled with different letters.

## 3. Results and Discussion

### 3.1. Determination of Available Lysine Groups

The OPA assay is suitable to quantify changes in the available lysine content of proteins because the complex is formed by lysine and OPA fluoresces [35]. Electrospinning brings the molecules in the fibers close together, and, at certain humidity levels, heat promotes glycation between the proteins and polysaccharides [26]. By analyzing the amount of free available lysine in the fibers, the progress of the conjugation between protein and polysaccharide during heating can be determined, since the ε-amino group of lysine residues is the primary source of amino groups during the Maillard reaction [39]. The concentration of free lysine groups was expressed as mg of lysine per g of the analyzed fibers (Figure 1A). For all the blends, only a slight change in lysine content was observed during heating. Of all the blends, only blends B and D showed significant differences, while all the other samples showed no significant changes after heating. Blend B showed a significant difference in lysine concentration between 6 h and 24 h (*p* < 0.05). Samples of mixture D after 0 h and 48 h differed significantly from the other heating times (*p* < 0.05). The samples also showed slight changes in protein content during heating; therefore, Figure 1 B shows the amount of available lysine in relation to the protein content of the sample, expressed as the percentage of lysine in the protein of the fibers. No significant differences were found between heating times, as shown in Figure 1B. The only significant differences observed were between the blends at the same heating time. Compared to the L-lysine concentration in potato protein isolate with a concentration of 6.0 ± 0.5 g/100 g protein, most of the fiber samples had a higher percentage of lysine in the protein content after electrospinning, with the exception of blend A after 6 h. The concentrations of lysine per protein content in the other blends were higher than in blend A. Table 1 shows that, after heating, the browning index of the fibers increased with an increase in heating time, and blend A, with the smallest fiber diameter, also had the highest browning index. Additionally, the protein ratios between mixture and fibers decreased with increased heat times. In a recent study, fibers with the same composition as mixture A were found to have the highest browning index [32], which is a typical indicator for an ongoing Maillard reaction between the proteins and the reducing sugars [20]. The other blends had significantly higher lysine concentrations in the protein than the fibers of blend A. Nevertheless, a nonsignificant decrease can be observed in the samples of blend B and D, as well as C, during heating. A decrease in free amino groups in the first hours of heating can be explained by the fact that the heating led to a rapid interaction between the polysaccharides and free amino groups [40]. In addition, prolonged heating times may result in the degradation and Amadori rearrangement of the initial products, which could affect the amount and conformation of the lysine residues [41]. Sometimes a decrease in free lysine groups can be observed at first, and then, with increasing heating time, an increase can be detected [31].

When the increase in exposure of new amino groups exceeds the loss due to glycation, an increase in total free amino groups can occur [40]. In a recent study on the conjugation of pea protein and maltodextrin via the Maillard reaction, it was suggested that higher exposure of amino groups might be caused by heating, due to protein unfolding, and might compensate for the decrease in free amino groups due to glycation [31]. A study with electrospun fibers from pea protein isolate and maltodextrin showed that furosines could already be detected in both the unheated fibers (1.1 mg/g protein) and the heated fibers, which had three times the content after 12 h at 65 °C and 75% RH [31]. Furosine, which can be formed in the first stage of the Maillard reaction from degradation of the Schiff base, can indicate an ongoing conjugation [42].

This could explain the slight decrease in free lysine groups after heating. Conjugation already takes place during electrospinning, before heating [31]. In conclusion, there are two distinct reaction steps: first, the unfolding of the protein with the increase in freely accessible amino groups, and second, the conjugation reaction via the Maillard reaction. It must be considered that the two reactions also partially occur simultaneously.

The fiber diameter of the samples may also play a role in the conjugation of electrospun fibers. In another study, it was shown that smaller diameters lead to a denser packing of the molecules, which promotes the conjugation [26]. Some researchers described the relationship between the ratio of maltodextrin and protein and its effect on glycation [20,43]. An excess of maltodextrin promotes the Maillard reaction, resulting in increasing glycation [18]. This suggests that, with increasing protein concentrations and constant amounts of maltodextrin in the fibers, glycation might decrease due to the lower ratio of maltodextrin to protein [20,43].

### 3.2. Determination of the Trypsin Inhibitor Activity

The trypsin inhibitor activity of potato protein–maltodextrin fibers was determined using azocasein as a protease substrate. The activity of an inhibitor of an enzyme is quantified by its inhibitory activity (IA). IA, expressed as the amount of trypsin inactivated by a certain quantity of sample, was calculated according to Equation (1) [38]. The amount of potato protein in the well of the microtiter plate of the dilutions from each sample was plotted against the absorption at 450 nm. A premise for the calculation of IA from the slope of the resulting line is that the decrease in absorbance is linear with the amount of trypsin (Figure 2A) [38]. This method was confirmed by a positive control, without potato protein (Figure 2B). The IA of all samples is shown in Figure 2C.

The IA of all blends decreased between 6 and 12 h to a nearly minimal level, which, in most cases, was significantly lower than that of the unheated fibers. However, after 24 h of heating, the IA increased again to almost the initial level. Measurements of the IA of pure potato protein isolate, that was neither electrospun with maltodextrin nor heated, showed a value of 517.8 ± 30.4 mg/g. Blend B showed only 87.8 ± 1.1% of the IA of the crude protein, and, thus, the strongest decrease in IA of all the different fiber blends directly after electrospinning (Table 2). Blend D showed an increase in IA of 104.1 ± 0.8% after spinning. Looking at mixture A, the IA seemed to decrease until 12 h of heating, then it increased again after 24 h. After 48 h, it reached the lowest value of 68.6 ± 6.5%. Mixtures B and D reached a higher IA after 48 h heating than the pure protein, with 105.3 ± 1.2% and 98.5 ± 2.3% respectively. Thus, the IA was significantly affected during heating (*p* < 0.001) and by the potato protein contents of the different blends (*p* < 0.001). In addition, the interaction between heating time and composition of blends was also significantly different (*p* < 0.001). In summary, the lowest significant values of IA in fiber blends were obtained between 6 and 12 h, with the exception of blend A after a heating time of 48 h. A clear trend for the development of IA after heating with increasing protein contents could be established from the given data. The higher the amount of potato protein isolate was in the blends, the higher the IA. However, the heating times showed no clear trend, because after 24 h the IA increased to higher levels than the unheated fibers, excepting the fibers of blend A and the pure potato protein powder.

Inhibition of proteinases by protein inhibitors occurs when access of the substrate to the catalytic center of the enzyme is hindered; usually, this is achieved by substrate-like binding of a peptide segment to the catalytic site [16]. In this process, substrate recognition sites are used by the inhibitor to achieve selectivity [16]. So-called canonical inhibitors have an exposed loop, which reacts with the catalytic residues of the proteinase in a similar way to the substrate [44]. In this process, side chains next to the loop interact with the enzyme, determining the specificity of the inhibitor for its enzymes [15].

In the case of trypsin specificity, lysine residues can also be involved. If these residues are then substituted, the inhibitory activity of trypsin can be influenced [16]. The structure of the loop of the serine protease inhibitor has not yet been fully described, but Meulenbroek et al. suggested that it possesses two reactive groups, and, thus, can bind two different proteases simultaneously [15]. The decrease in IA could be explained by the fact that the lysine residues participating in the inhibition process are blocked by the Maillard reaction. Cross-linking and steric hindrance occur; thus, the inhibition is hindered [45,46]. The IA values of the samples are influenced by which lysine groups participate in the conjugation and whether the conjugation is near the reactive sites of the inhibitor [46]. The active sites of the potato serine protease inhibitor (PSPI) have not yet been clearly identified. There are several suggestions for active sites. One possibility for the reactive-site loop of PSPI is located around Lys95 [15]. Meulenbroek et al. concluded from the fact that a large positively charged amino acid is best suited to contact the active site of trypsin that the Lys95 loop is the loop that interacts with trypsin [15]. Billinger et al. suggested that most of the possible conjugation sites of PSPI probably have no effect on inhibition [46]. However, since Lys95 is also a possible conjugation site, conjugation at this position could, of course, affect the IA. The trypsin inhibitory activity of potato trypsin inhibitors decreased slightly at low heating temperatures; however, when the temperature increased from 60 to 70 °C, the inhibitory activity decreased sharply, with a loss of 79% [47]. This study showed that the structure of potato trypsin inhibitors unfolded when the proportion of β-sheet and β-turn decreased between 70 and 100 °C, while the proportion of α-helix and random coil increased [47].

Since IA was already expressed in relation to the protein content of the samples, it could be concluded that IA would either remain constant between the different blends, or that the ratio between protein and maltodextrin would have an influence on the glycation, blocking more lysine, thus influencing the IA. In the present study, the lowest protein content (blend A) had the highest decrease in IA, and, in a recent study, the same composition of heated fibers (blend A) had the highest browning index, as an indicator of the initial and intermediate products of the Maillard reactions [32]. However, since the calculated IA is difficult to interpret in terms of glycation, due to the mostly nonsignificant results of the OPA assay, the reduction in IA cannot be directly attributed to glycation of the ε-amino groups of lysine. Further studies should be performed to confirm the mechanistic relationship between glycoconjugation and decreases in IA due to conjugation of the inhibitor.

### 3.3. FTIR

By recording an FTIR spectrum, it is possible to obtain information about the chemical structure of the examined material. By comparing the spectra of the unheated and heated potato protein maltodextrin fibers, it is possible to draw conclusions about the effects of glycation during the heating of the fibers. When examining the spectra of proteins, the two peaks identified at around 1636 cm^−1^ and 1549 cm^−1^ (which are the amide I and amide II regions) were important, because they were characteristic of the structure of a protein [48]. In this study, the absorbance peaks of the regions of the amide I and amide II bands were observed. The FTIR spectra of blends A, B, C, and D are shown in Figure 3.

In the potato protein, peaks were found at 1634 cm^−1^ (amide I) and 1549 cm^−1^ (amide II). The amide II band, for secondary amides, is due to the coupling of N–H bending and C–N stretching, and normally appears at 1560–1530 cm^−1^ [49]. In blend A, a shift in the amide I region, from 1640 cm^−1^ at 0 h to 1645 cm^−1^ after 12 h heating, was observed, but after 48 h of heating, it remained at a wavenumber of 1645 cm^−1^.

In none of the samples of blend A could any amide II bands be directly detected, probably due to the low protein content of the fibers. Only after deconvolution of the peaks could a shift from 1531 cm^−1^ in the unheated fiber to 1540 cm^−1^ after a thermal treatment of 48 h be detected.

In a recent study with maltodextrin-WPI fibers, a shift to smaller wavelengths was observed [30]. The wavenumber of the amide I region of fibers (blend B, 0 h) was 1645 cm^−1^; then, it shifted to 1643 cm^−1^ after 12 h of heating. After 48 h of heating, the amide I region was again 1645 cm^−1^. In the amide II region, a shift from 1539 cm^−1^ to 1536 cm^−1^ could be observed after 48 h. The amide I region in the fibers of blend C (0 h) was found at 1644 cm^−1^; then, it shifted to 1647 cm^−1^ after 6 h, and after 48 h, it was again at 1643 cm^−1^. The amide II was first at 1536 cm^−1^; then, it shifted after 24 h to 1533 cm^−1^. In sample D, the amide I region changed from 1643 cm^−1^ to 1641 cm^−1^ after 48 h. For the amide II band, only a slight shift from 1537 cm^−1^ to 1532 cm^−1^ after 24 h was analyzed.

The amide II peak is caused by stretching of the C-N and C-C groups, and in-plane N-H bending [49]. The amide I band can be expected to be explained by the spectral overlap of the in-plane bend of N-H and the C=N linkage associated with the C=O group [50]. Group frequencies, such as those of the amide I and amide II groups, can be influenced by conjugation, for example [49]. Various authors have presented a shift in the bands of amide I and amide II to smaller wavenumbers, which is caused by glycation; the formation of Schiff bases and the consumption of amino groups during the Maillard reaction cause this shift [51]. Liu and Zhong (2012) observed a shift of the absorption bands from 1640 cm^−1^ to 1634 cm^−1^ (amide I), and from 1596 cm^−1^ to 1581 cm^−1^ (amide II) after glycation of WPI with maltodextrin. Kosaraju et al. reported a shift of the amide I band from 1654 cm^−1^ to 1637 cm^−1^ in chitosan–glucose conjugates [51]. After glycation, other authors observed, in sweet potato protein, that the amide I and II stretching bands were found at 1672 and 1550 cm^−1^, respectively [52]. Hydrogen bonding is also among the most important intermolecular effects that may be responsible for the shift of peaks [49]. Because the observed shifts in the amide I and amide II regions were not as large as those described in the literature, it was concluded that conjugation in the produced fibers could be only detected by FTIR in tendency. One reason for this was the partially low protein content in the fibers; 5–20% of the concentration of maltodextrin.

The PCA (Figure 4) of all FTIR spectra showed that there were no major structural differences between the spectra of the different fibers (Figure 4A) or the different heating times (Figure 4B), except for the spectra of fiber samples that were not heated. Here, the symbols that are close to each other have similar spectra, and, thus, structures, while those that are far apart are dissimilar. In Figure 4A, the PCA of the different mixtures shows that samples of the mixture D were close to each other in the center of the PCA and slightly different from mixtures B and C, except for the marked outliers (B 0 h and C 6 h). The fibers of blend A were located farther from the center and differed from B and C.

The PCA results showed that the sample A 6 h and A 24 h had little difference. Blend B represented a group of spectra with some similarity of spectra, and, thus, glycation degree with blend D. Figure 4B shows the PCA score plot of the different heating times. The spectra of the heated fibers (12, 24, and 48 h) were close to the center of the plot, indicating that they had similar structures and, thus, glycation degrees. However, three spectra were marked as outliers. In other studies, FTIR was able to discriminate between glycated and non-glycated, e.g., sodium caseinate, when the data were analyzed by multivariate statistical methods such as PCA [53].

### 3.4. Key Insights

In summary, Figure 5 shows the most commonly observed effects of electrospinning and subsequent thermal treatment on the IA of potato protein as a function of its concentration. The core hypothesis of this work was that electrospinning should provide a larger surface area for the potato protein to facilitate glycation of the potato protein with maltodextrin via the Maillard reaction. Using the browning index as an indirect indicator of the Maillard reaction, it was shown that the Maillard reaction proceeded, and a higher browning index was obtained at smaller fiber diameters. However, significant browning was avoided, as the browning index was less than 25 [20]. Both the determination of free lysine and the FTIR spectra only tended to indicate an increasing degree of glycation in the different fiber samples. However, it appeared that trypsin inhibition was possible, due to the thermal treatment, and the lowest fiber diameter had the lowest IA after a thermal treatment of 12 h.

## 4. Conclusions

Natural protease inhibitors derived from potatoes are capable of inhibiting enzymes such as trypsin. The analytical results of IA obtained by the azocasein assay were suitable for the detection of IA in complex matrices such as electrospun potato protein–maltodextrin fibers. These fibers were heated to generate polysaccharide–protein conjugates. This resulted in a slight decrease in lysine content in the produced fibers after short-term heating, which is likely attributable to the degree of glycation of each sample. As hypothesized, the IA of the fibers showed significant changes at different protein contents and heating times. However, no linear decrease in IA could be detected. When 5% potato protein was added to the mixture, a maximum decrease in IA to 74% was observed at the lowest fiber diameter (1.4 µm) after 12 h. The relationship between the number of free lysine groups and the inhibitory effect on trypsin could not be directly demonstrated, possibly due to the unfolding and structural change of the potato protein during heating. Further studies are needed to determine the degree of glycation in the different fiber samples.

## Figures and Tables

**Figure 1 foods-11-00918-f001:**
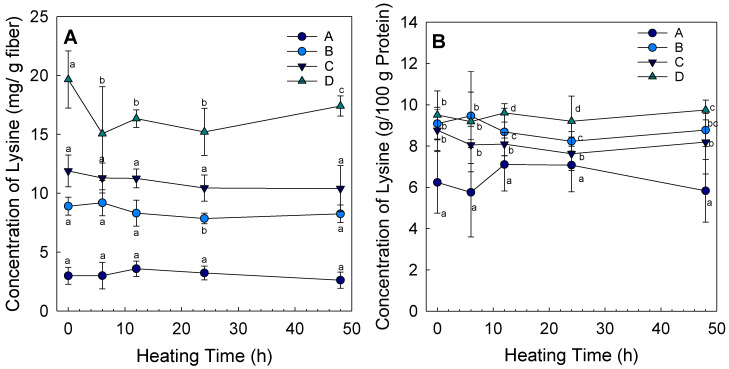
(**A**) Average concentration of lysine (mg/g fiber) and (**B**) concentration of lysine in the protein content over heating times for the different ratios (MD DE 2:MD DE 21:PP)(A (80:10:5), B (80:10:10), C (80:10:15), and D (80:10:20)) with standard deviation (different letters indicate significant differences (**A**) between heating times at the same blend and (**B**) between blends at the same heating time (*p* < 0.05)).

**Figure 2 foods-11-00918-f002:**
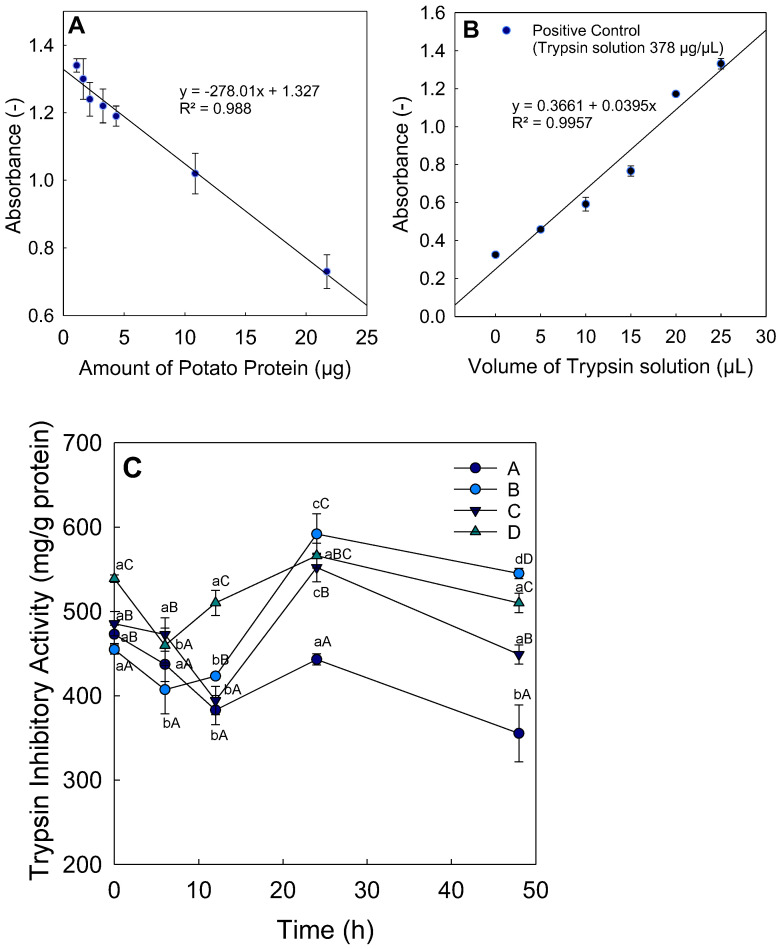
(**A**) Measurement of the trypsin inhibitory effect of pure potato protein (absorbance measurement at 450 nm over the quantity of potato protein in the assay), (**B**) positive control without potato protein, and (**C**) trypsin inhibitory activity of all fibers (MD DE 2:MD DE 21:PP—A (80:10:5), B (80:10:10), C (80:10:15), and D (80:10:20)) vs. heating time (different letters indicate significant differences between heating times (lower case) or blends (upper case) (*p* < 0.05)).

**Figure 3 foods-11-00918-f003:**
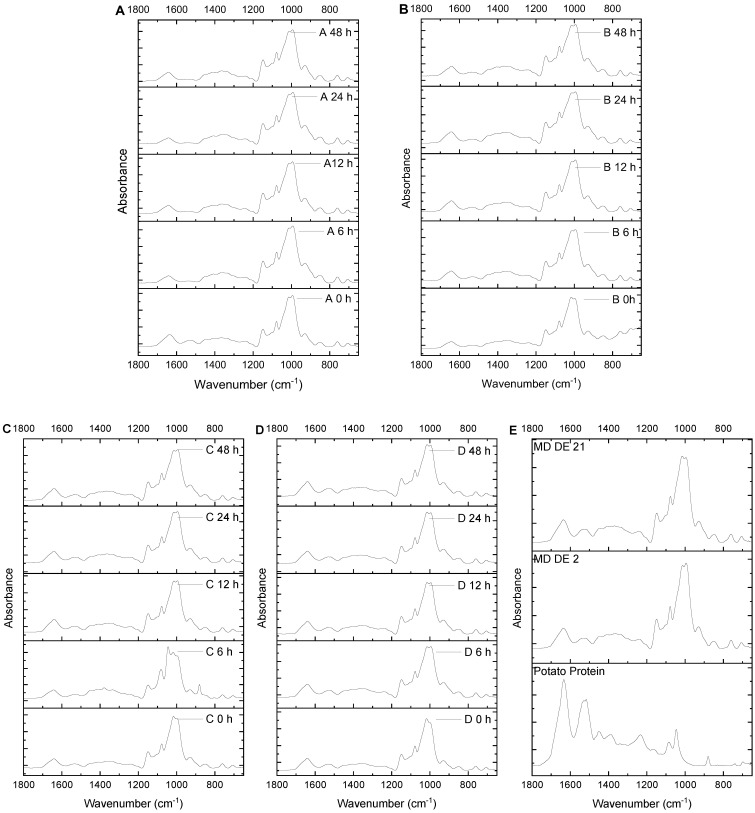
FTIR spectra (700−1800 cm^−1^) of all fibers with different ratios (MD DE 2:MD DE 21:PP)—(**A**) (80:10:5), (**B**) (80:10:10), (**C**) (80:10:15), (**D**) (80:10:20), and (**E**) maltodextrins DE 21, DE 2, and potato protein.

**Figure 4 foods-11-00918-f004:**
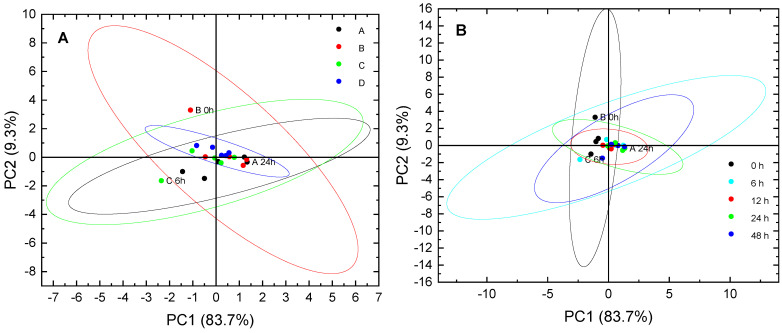
PCA of the different FTIR spectra of (**A**) different blends (MD DE 2:MD DE 21:PP—A (80:10:5), B (80:10:10), C (80:10:15), and D (80:10:20)) and (**B**) different heating times. (Labelled spectra are outliers, and the confidence ellipsoids represent in each case a probability of 95%).

**Figure 5 foods-11-00918-f005:**
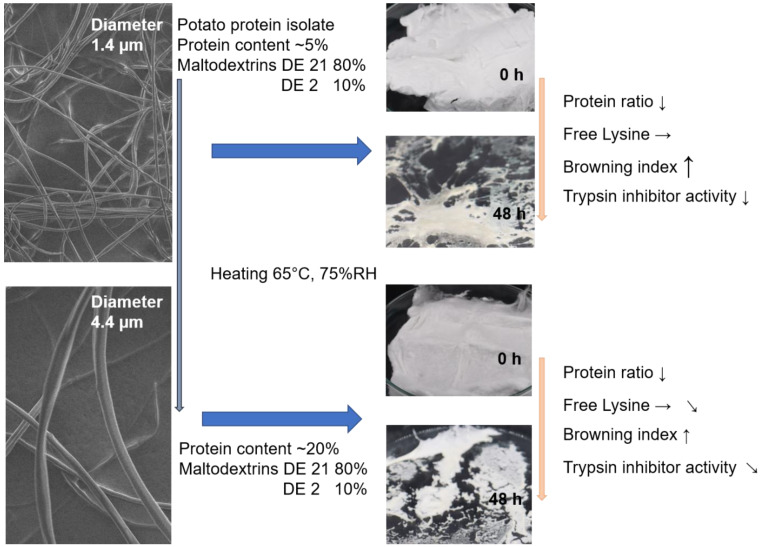
Schematic illustration of the study with respect to the addition of different concentrations of potato protein and the effects of electrospinning and thermal treatment on trypsin inhibitor activity.

**Table 1 foods-11-00918-t001:** Browning indexes at different heating times, diameters of the fibers, and protein ratios between mixture and fibers (different letters indicate significant differences between heating times in the same blend (lower case) and between blends at the same heating time (upper case) (*p* < 0.05)).

Blend	Heating Time	Browning Index (-)	Diameter (µm)	Protein Ratio Mixture/Fiber
A	0 h	1.23 ± 0.06 ^aA^	1.43 ± 0.75 ^a^	0.96 ± 0.001 ^aA^
	12 h	7.14 ± 0.48 ^bA^		1.01 ± 0.003 ^bA^
	48 h	8.37 ± 1.02 ^bcA^		0.90 ± 0.013 ^cA^
B	0 h	1.49 ± 0.02 ^aB^	2.31 ± 0.69 ^b^	0.98 ± 0.002 ^aB^
	12 h	5.24 ± 0.66 ^bB^		0.96 ± 0.001 ^bB^
	48 h	7.76 ± 0.09 ^cA^		0.94 ± 0.028 ^bB^
C	0 h	1.94 ± 0.20 ^aC^	2.51 ± 0.69 ^b^	0.91 ± 0.001 ^aC^
	12 h	3.98 ± 1.23 ^b^		0.93 ± 0.002 ^bC^
	48 h	5.95 ± 068 ^cB^		0.85 ± 0.015 ^cC^
D	0 h	2.44 ± 0.06 ^aD^	4.35 ± 1.58 ^c^	1.03 ± 0.014 ^aD^
	12 h	5.10 ± 1.80 ^bC^		0.85 ± 0.015 ^bD^
	48 h	7.69 ± 1.44 ^cA^		0.89 ± 0.008 ^cD^

**Table 2 foods-11-00918-t002:** Means and standard deviations of the ratio of IA (mg trypsin inhibited per g protein) of the potato protein incorporated in the spun and heated protein–maltodextrin fibers, and the IA of potato protein isolate powder.

Heating Time (h)	Trypsin Inhibitor Activity (%) ^1^				
	Mean	A (80:10:5)	B (80:10:10)	C (80:10:15)	D (80:10:20)
0	94.2 ± 6.5 ^a^	93.1 ± 2.1 ^bA^	87.8 ± 1.1 ^bA^	93.8 ± 2.8 ^bA^	104.1 ± 0.8 ^bcB^
6	85.8 ± 6.2 ^b^	84.4 ± 3.9 ^bA^	78.6 ± 5.6 ^aB^	91.3 ± 3.8 ^bC^	88.8 ± 4.0 ^aAC^
12	82.6 ± 10.3 ^c^	74.0 ± 3.3 ^aA^	81.7 ± 0.5 ^aB^	76.1 ± 3.2 ^aA^	98.5 ± 2.9 ^bC^
24	104.0 ± 11.3 ^d^	85.6 ± 1.3 ^bA^	114.3 ± 4.6 ^dBD^	106.6 ± 3.3 ^cC^	109.3 ± 2.8 ^cD^
48	89.8 ± 14.8 ^e^	68.6 ± 6.5 ^aA^	105.3 ± 1.2 ^cB^	86.7 ± 2.2 ^bC^	98.5 ± 2.2 ^bD^
Blend Mean		80.8 ± 9.2 ^A^	93.5 ± 15.5 ^B^	90.9 ± 11.1 ^C^	99.8 ± 7.6 ^D^
Heating time	<0.001				
Blend	<0.001				
Time × blend	<0.001				

^1^ Different letters indicate significant differences (*p* < 0.05) between the means within each column (lower case letters) and within each row (upper case letters).

## Data Availability

The data presented in this study are available on request from the corresponding author.

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
