# Peer review of "Effects of Electrospun Potato Protein–Maltodextrin Mixtures and Thermal Glycation on Trypsin Inhibitor Activity"

_foods, 2022, doi:10.3390/foods11070918_

Round 1

Reviewer 1 Report

General comments

The submitted paper is aimed at investigating the trypsin inhibitory activity of potato protein as a result of the electrospinning and glycation with maltodextrin via the Maillard reaction.

The topic of the present work is appealing and worthy of investigation. Moreover, it well fits the aim and topic of Foods. However, some revisions have to be applied. A description of the obtained electrospun products has to be added, introduction a scanning electron microscopy (SEM) investigation of the samples before and after grounding and before and after the applied thermal treatments.

A deep English grammar and language revision is strongly recommended.

More details and specific remarks and suggestions are reported below point by point.

Abstract

  • The Abstract is too long and has to be summarised in order to make it more incisive, highlighting the main outputs.
  • A contextualization has to be added as incipit.
  • The acronym DE has to be specified the first time it is used, as well as FTIR

Keywords

The chosen keywords (i.e. glycoconjugates; potato protein; maltodextrin; trypsin inhibitor; free lysine groups) do not completely cover the manuscript content. Please add some ones about the technique, the product and the application.

Introduction

- The Introduction section is well organised, but the originality of the present work with respect to the previous reports about the same topic, particularly compared to previous papers by the same Authors, has to be evidenced, at the end of the Introduction section.

- The incipit “Protein isolates are used in the production of a wide range of foods” has to be supported with proper suitable literature references, in order to evidence the need to reuse agro-food waste by-products, including “Influence of diverse natural biopolymers on the physicochemical characteristics of borage seed oil-peppermint oil loaded W/O/W nanoemulsions entrapped with lycopene. Nanotechnology, 32(50) (2021), 505302.”.

- More details about the electrospinning process have to be added, as well as some examples of its application in the food sector, reporting the related references, such as “Encapsulation of bioactive compounds from aloe vera agrowastes in electrospun poly (ethylene oxide) nanofibers. Polymers, 12(6) (2020), 1323.”.

  1. Materials and Methods

2.2 Preparation of electrospun fibers and their thermal treatment

- Even if reported elsewhere, the electrospinning process parameters (applied voltage, needle-target distance, flow rate) and the environment conditions (temperature and relative humidity) have to be specified.

2.6 FTIR-analysis

The resolution has to be specified.

  1. Results and Discussion

This section is well described and discussed, also comparing the obtained data witht the literature results.

However, the Authors did not report any information about the obtained fibers. Even if they ground the electrospun product, it is important to know if it was possible to obtain fibers or not. Thus, SEM micrographs of the samples before and after grounding and before and after the thermal treatments have to be added, described and discussed. 

 3.3 FTIR

  • The following sentences “By recording a FTIR spectrum, it is possible to obtain information about the chemical structure of the examined material. By comparing the spectra of the unheated and heated potato protein maltodextrin fibers, it is possible to draw conclusions about the effects of glycation during the heating of the fibers” have to be supported with proper references.
  1. Conclusions

Some numerical data have to be added.

Author Response

We appreciate the time taken by reviewer to go through our manuscript, and in response, have made changes to the manuscript. Responses to comments are shown below and were marked in blue. Changes in the actual manuscript have been marked in red. The line numbers refer to the revised and attached Word document.

Reviewer 2 Report

  • The whole idea (hypothesis) of the work is that electrospinning shall offer a higher surface area for the potato protein to facilitate potato protein glycation with maltodextrin via the Maillard reaction. The resulting electrospun product shall cause a decrease of the trypsin inhibitory activity as a result of the large surface area and mild glycation conditions.
  • Under introduction on line 49, the correct term is physicochemical properties rather than technological properties when referring to solubility, foaming, or emulsification.
  • On line 51, what is meant by functional property? Please specify. Is this biological, pharmacological, or what exactly?
  • Line 105 – It is better to use "were subjected to" instead of was applied …….electrospun fibers were subjected to a thermal treatment…
  • Although referred to reference 27, one of the authors' previous similar works, more details regarding the blends and their rations must be provided under methodology.
  • On line 201, it is stated that "The concentrations of lysine per protein in the other blends are higher than in blend A." More elaboration on the rationale and reason should be provided.
  • Figure 1 (A & B) are not clear nor explained under methodology. What are small a and b denoting for? The reader should be able to easily understand and follow.
  • Under FTIR analysis, it is more accurate to use term bands instead of peaks.
  • The central hypothesis of this paper, as stated above, was to demonstrate that electrospinning shall offer a higher surface area for the potato protein to facilitate potato protein glycation with maltodextrin via the Maillard reaction. This study is considered incomplete and shallow without providing information about the degree of glycation in the different fiber samples. The authors must provide what supports the hypothesis by providing this data in this paper as the whole hypothesis of protease inhibition is dependent on this.

Author Response

Please see the pdf file below.

Reviewer 3 Report

After careful reviewing the manuscript titled “Effects of Electrospinning of Potato Protein-Maltodextrin Mixtures and Thermal Glycation on Trypsin Inhibitor Activity”. There is not much to be improved as manuscript is well written and presented. Language is also suitable. However, I have few suggestions for authors which they should correspond before final publication.

  1. Graphical abstract should be provided as Figure 1 in the manuscript which clearly describe purpose, methodology, and main findings of the study. Following articles can be viewed as reference for preparation of graphical abstract having all information;

https://doi.org/10.1021/acsanm.0c01562; https://doi.org/10.1038/s41598-019-49132-x;

  1. X-Axis of FTIR spectra should be reversed (for example 1800-800, instead of 800-1800)
  2. Is the wavenumber nm-1 or cm-1. Please confirm

Author Response

Please see the pdf file below.

Round 2

Reviewer 1 Report

General comments

The Authors have followed almost all the Referees’suggestions.

However, some minor revisions have to be applied.

Abstract

  • A contextualization has not been added as requested in the previous review.

Introduction

- As already requested in the previous review, the originality of the present work with respect to the previous reports about the same topic, particularly compared to previous papers by the same Authors, has to be evidenced, at the end of the Introduction section.

  1. Materials and Methods

2.2 Preparation of electrospun fibers and their thermal treatment

- For the electrospinning process, an applied voltage of 64 V is too low…Please check it.

- The flow rate has to be specified.

  1. Results and Discussion

The Authors have improved this section. The Reviewer asked to introduce the microstructural results regarding the obtained fibers, since the work is based on the hypothesis that the electrospun fibers can provide a larger surface area for the potato protein to facilitate glycation of the potato protein with maltodextrin via the Maillard reaction

Author Response

Please see below the pdf file.

Reviewer 2 Report

  • As indicated in my previous comments, on line 45, the correct term is physicochemical properties rather than technological properties when referring to solubility, foaming, or emulsification.
  • Out of curiosity, what is the reason for not using LC/ms for accurate quantification of the degree of protein glycation

Author Response

Please see below the pdf file.
